# The Clinical and Psychosocial Outcomes for Women Who Received Unexpected Clinically Actionable Germline Information Identified through Research: An Exploratory Sequential Mixed-Methods Comparative Study

**DOI:** 10.3390/jpm12071112

**Published:** 2022-07-07

**Authors:** Laura E. Forrest, Rowan Forbes Shepherd, Erin Tutty, Angela Pearce, Ian Campbell, Lisa Devereux, Alison H. Trainer, Paul A. James, Mary-Anne Young

**Affiliations:** 1Parkville Familial Cancer Centre, Peter MacCallum Cancer Centre, Melbourne, VIC 3000, Australia; laura.forrest@petermac.org (L.E.F.); rowan.forbesshepherd@petermac.org (R.F.S.); erin.tutty@mcri.edu.au (E.T.); alison.trainer@petermac.org (A.H.T.); paul.james@petermac.org (P.A.J.); 2Sir Peter MacCallum Department of Oncology, The University of Melbourne, Melbourne, VIC 3010, Australia; ian.campbell@petermac.org (I.C.); lisa.devereux@petermac.org (L.D.); 3Kinghorn Centre for Clinical Genomics, Garvan Institute of Medical Research, Darlinghurst, NSW 2010, Australia; a.pearce@garvan.org.au; 4Cancer Genetics Laboratory, Peter MacCallum Cancer Centre, Melbourne, VIC 3000, Australia; 5**Life**pool, Peter MacCallum Cancer Centre, Melbourne, VIC 3000, Australia

**Keywords:** returning research results, clinically actionable genetic information, mixed-methods, psychosocial outcomes, clinical outcomes

## Abstract

Background Research identifying and returning clinically actionable germline variants offer a new avenue of access to genetic information. The psychosocial and clinical outcomes for women who have received this ‘genome-first care’ delivering hereditary breast and ovarian cancer risk information outside of clinical genetics services are unknown. Methods: An exploratory sequential mixed-methods case-control study compared outcomes between women who did (cases; group 1) and did not (controls; group 2) receive clinically actionable genetic information from a research cohort in Victoria, Australia. Participants completed an online survey examining cancer risk perception and worry, and group 1 also completed distress and adaptation measures. Group 1 participants subsequently completed a semi structured interview. Results: Forty-five participants (group 1) and 96 (group 2) completed the online survey, and 31 group 1 participants were interviewed. There were no demographic differences between groups 1 and 2, although more of group 1 participants had children (*p* = 0.03). Group 1 reported significantly higher breast cancer risk perception (*p* < 0.001) compared to group 2, and higher cancer worry than group 2 (*p* < 0.001). Some group 1 participants described how receiving their genetic information heightened their cancer risk perception and exacerbated their cancer worry while waiting for risk-reducing surgery. Group 1 participants reported a MICRA mean score of 27.4 (SD 11.8, range 9–56; possible range 0–95), and an adaptation score of 2.9 (SD = 1.1). Conclusion: There were no adverse psychological outcomes amongst women who received clinically actionable germline information through a model of ‘genome-first’ care compared to those who did not. These findings support the return of clinically actionable research results to research participants.

## 1. Introduction

Access to genetic information has increased as direct-to-consumer models, research initiatives, and private companies offer germline testing to anyone irrespective of their underlying risk. These pathways promote a significant departure from the model of care offered by clinical genetics services (CGS), where access is driven by personal and family history of disease (e.g., breast cancer). CGS have traditionally been the largest provider of genetic information and offer genetic testing nested within the practice of genetic counselling, where pre-and post-test appointments facilitate preparation for, and delivery of, genetic information [1]. The CGS model arose due to the risk of harm to individuals and families receiving genetic information without support [2,3]. Genetic counselling fosters informed consent, promotes empowerment and adaptation, facilitates family communication, and encourages the uptake of cancer risk management strategies [4,5,6].

In contrast to traditional CGS models, the ‘genome-first’ approach takes a more liberal approach to cancer risk assessment, beginning with genotype rather than phenotype [7,8]. For example, taking a genome-first approach to determining familial breast cancer risk means using the genomic investigation as a first-line assessment, rather than starting with personal and/or family history of breast cancer [8]. In research settings, genome-first approaches identify participants with clinically actionable germline information that is “*independent of any clinical parameters*” [7]. Genome-first approaches may reach far more of the population, likely before the onset of disease, and provide more efficient risk estimation than CGS models [9]. This paradigm nonetheless has implications for disease penetrance estimates, recommendations for risk management, and identification of at-risk family members. Research participants who receive germline information therefore ideally require referral to CGS for genetic counselling [7,10]. The experiences of research participants and outcomes for populations who receive genome-first care, however, are not well described and may have implications for genetic counselling practices.

The lived experiences of many women with a family history of breast cancer are a primary source of knowledge on how to live with cancer and cancer risk, and greatly influence perceptions of cancer risk and risk management [11,12,13,14,15]. An elevated perception of risk may negatively impact some women leading them to adopt inappropriate risk management strategies [12]. In contrast, many research participants may have limited or negligible lived experience of cancer, and their risk perception may differ from those who have a strong family history of disease. As a result, their responses to receiving this genetic information, decisions about cancer risk management, and family communication may be different to the population well characterised in the literature.

In Australia, clinically actionable germline pathogenic variants (PVs) identified through research are often returned to participants via a genetic counsellor-mediated pathway [16,17]. The notification process consists of (1) a letter notifying participants that information relevant to their health has been identified, and (2) follow-up by a genetic counsellor (GC) via a telephone genetic counselling (TGC) service [17]. This process allows research participants to obtain information and support enabling them to make an informed decision about whether or not they wish to receive their results. Participants who receive their research results are subsequently referred to CGS for ongoing management including confirmatory genetic testing and genetic counselling [16,17].

It is important to understand the clinical and psychosocial outcomes for individuals who receive clinically-actionable genetic information through research to inform models of care and service delivery for individuals who have not had formal pre-test genetic counselling and may not have lived experience of the condition [10]. Further, the increasing interest in population genomic testing underlines the need to develop an evidence-based policy for best practice in returning results. Recognising this, this study aimed to examine the clinical and psychosocial outcomes for women in a population-based cohort (**life**pool) that returned clinically actionable research results for hereditary breast and ovarian cancer (HBOC) [18] by:(1)comparing women’s cancer worry after receiving or not receiving clinically actionable genetic information from **life**pool as measured by the cancer worry scale; and(2)exploring outcomes including cancer risk perception, uptake of cancer risk management strategies, the experience of receiving and adapting to the information, and how the information is communicated to at-risk family members.

## 2. Materials and Methods

We used a sequential, comparative mixed method research design consisting of qualitative interviews and an online survey using case and control participant groups [19]. This approach allowed us to identify differences in key psychosocial outcomes (risk perception and cancer worry) between women who did and did not receive genetic information from **life**pool, while also qualitatively exploring women’s experiences of receiving this information. Survey and interview data were analysed separately and integrated at the interpretation stage. A separate in-depth qualitative study of women’s experiences has been previously reported [20].

### 2.1. Setting and Sample Population

Participants were sampled from **life**pool, a research cohort recruited from a population-based breast cancer screening program, described elsewhere [18,20]. At the commencement of this study, 55,000 women had been recruited to **life**pool, 17,000 had provided a DNA sample for germline sequencing with a custom 11-gene HBOC panel, and 75 women had been notified that a clinically actionable germline PV had been identified in their sample. Notification occurred by a two-step pathway with participants receiving a letter from **life**pool, then followed up by a genetic counsellor from the Parkville Familial Cancer Centre at the Peter MacCallum Cancer Centre who provided telephone genetic counselling [18].

### 2.2. Participants, Sampling, and Recruitment

Participants were recruited as cases or controls in a 1:2 ratio. Cases (group 1 participants) were eligible if they had been identified with a clinically actionable germline PV in genes *BRCA1, BRCA2, PALB2, or ATM* and notified of this information. Controls (group 2 participants) were also recruited from the **life**pool population and were age- and education-matched to group 1. Group 2 women had provided a DNA sample to **life**pool, which had been sequenced with no clinically actionable PVs identified and were not notified of this information. All over recruitment numbers of 48 cases and 96 controls were based on an 80% probability of identifying a difference at a 5% significance level (two-sided) with a moderate effect size (0.5) using the primary outcome of cancer worry. Eligible participants were invited from July 2019 to September 2020 by email, which included a URL to access online participant information and consent. Group 1 participants had been notified about the clinically actionable germline PV by **life**pool between June 2014 and May 2020.

### 2.3. Data Collection

After providing consent participants accessed an online survey hosted using REDCap software at the Peter MacCallum Cancer Centre [21,22]. The survey for group 1 included six domains and the survey for group 2 included domains 1 and 3 (see Appendix A for the surveys).

Domain 1. DemographicsDomain 2. Genetic testing and risk managementDomain 3. Cancer risk perception and cancer worryDomain 4. Impact of cancer risk assessmentDomain 5. Psychological adaptationDomain 6. Family communication

### 2.4. Measures

Items in domains 2 and 6 were purpose designed. Affective and cognitive risk perception measures were used to assess the risk perception of developing breast and ovarian cancer (domain 3) [23,24].

The ***Cancer Worry Scale (CWS*)** is an eight-item questionnaire that measures concerns about developing cancer (again), worries about family, and future surgery. Higher CWS scores indicate higher worry. The CWS was used to assess concerns about developing breast and/or ovarian cancer (again) (domain 3) [25].

The ***Multidimensional Impact of Cancer Risk Assessment (MICRA)*** is a 25-item questionnaire with three subscales: distress, positive thinking, and uncertainty [26]. The MICRA measures sub-clinical symptoms of individuals who receive genetic test results. Higher MICRA scores indicate greater distress, positive thinking, or uncertainty, respectively. The MICRA was used to assess psychological responses after receiving genetic information from **life**pool (domain 4).

The ***Psychological Adaptation Scale (PAS)*** is a 15-item questionnaire that measures adaptation to a chronic condition or disease risk. Higher PAS scores indicate greater adaptation to the condition. The PAS was used to measure the psychological adaptation of women who had received genetic information from **life**pool (domain 5) [4].

For the present sample, the Cronbach’s alpha for each of these scales were: CWS = 0.80, MICRA = 0.82, and the PAS = 0.96.

### 2.5. Interviews

After completion of the survey, a subset of purposively sampled women from group 1 were interviewed (authors RFS and ET) via telephone or in person. Interviews were guided by a semi-structured schedule that covered experiences of notification, perceived value of and adaptation to notification, cancer risk perception, risk management, and family communication [20]. Interviews were audio recorded and transcribed verbatim. Transcripts were quality controlled, de-identified, and pseudonyms assigned before analysis.

### 2.6. Data Analysis

#### 2.6.1. Quantitative

Demographics were described per group and compared between groups using chi-squared tests and independent samples *t*-tests. Logistic regression modelling was used to examine predictors of uptake of risk-reducing surgery by group 1. Risk perception and cancer worry were compared between groups 1 and 2 using independent samples *t*-tests. In the absence of evidence-based guidelines for the interpretation of between-group differences in the CWS and risk perception scales, the Cohen’s *d* effect size was calculated to characterise the size of observed differences (small = 0.2; medium = 0.5; and large = 0.8) [27]. Responses to the CWS scale were dichotomised into low (≤13), and high cancer worry (≥14) and groups were compared using chi-squared testing [25]. Data were analysed using Stata BE/17. *p* values < 0.05 were considered statistically significant.

#### 2.6.2. Qualitative

Team-based, codebook thematic analysis used to generate qualitative findings [28,29], is described previously [20]. Recruitment ceased after reaching requisite informational power to answer the research questions. QSR International NVivo v12.6.0 software (Massachusetts, USA) supported data management and analysis. Participant quotes are used to illustrate the findings with a pseudonym, age at interview in years, and the gene within which the germline PV was identified in parentheses.

## 3. Results

Sixty-eight women in Group 1 were invited to participate, 10 declined, 8 were uncontactable at follow-up, and 4 agreed to interview only, resulting in 46 survey respondents. One participant was subsequently deemed ineligible due to a partial response, resulting in 45 completed surveys (response rate, 66%). Of these respondents, 31 subsequently participated in an interview (mean age 62.5 years) on average 2.3 years (range 0.3–5.1 years) after result notification. Most interview participants had a *BRCA2* PV (*n* = 22, 71%), detailed demographics have been published [20].

One hundred and fifty-seven women in group 2 were invited and 115 consented. Twelve did not complete the survey, and a further 7 were subsequently deemed ineligible, resulting in 96 respondents (response rate 61%).

### 3.1. Demographics

There were no differences between groups 1 and 2 in terms of age, education, income, and relationship status. A greater proportion of group 1 had children (93.3%) compared to group 2 (79.2%) (*p* = 0.03). All women in group 2 were unaffected by cancer, whereas one third of group 1 had a personal cancer diagnosis (Table 1).

### 3.2. Uptake of Confirmatory Genetic Testing and Cancer Risk Management Strategies

Ninety-six percent of group 1 participants (43/45) had undergone confirmatory genetic testing through a CGS, and 76% (34/45) engaged in breast cancer risk management strategies (Table 2). Participants who had a risk-reducing mastectomy (RRM) were younger (mean 54.2 years, SD 9.2) than participants who had not (mean 66.9 years, SD 6.8; *p* < 0.001). Multivariate analyses showed no independent predictors (time since genetic testing (years), tertiary education, income, relationship status, or age) were associated with uptake of RRM. Whereas greater time since genetic testing was associated with uptake of bilateral salpingo-oophorectomy (BSO) (OR 3.7, *p* = 0.02, 95% CI 1.2–11.4) and uptake of tamoxifen (OR 1.8, *p* = 0.05, 95% CI 1.0–3.1) when controlling for tertiary education, income, relationship status, and age. There were no independent predictors for uptake of breast screening.

While nearly all women had completed confirmatory testing, some described the waiting period between being notified by lifepool and attending their clinical genetic service appointment as a distressing time.

*I had to wait nine weeks. I think that was probably the hardest part for me… if people have been kind enough to give their time and their energy to research, I think you probably could expedite that process for them in a way. Rather than just let them wait in the system*. (Bridget, 54 years, *BRCA2*)

Further, waiting to confirm that they were at high risk of cancer prompted feelings of uncertainty and anxiety, which persisted until they received their clinically confirmed results at an FCC appointment.

*Before I told anybody, I had to have a second test done, which took maybe six weeks and I think that was probably the worst time of anxiety for me, knowing, but not saying anything, because I wanted to be a hundred percent sure. That [test] actually took a lot longer and was a lot more badly organised. It was an extra month past the time that it was received by [hospital], to when I was actually advised, so I don’t know why that delay was there? I think that could have been handled better*.(Evie, 51 years, *BRCA2*)

### 3.3. Decision-Making and Motivation Regarding Cancer Risk Management

Women’s attitudes and subsequent decision making with regards to risk management were informed by their risk perception, attitudes towards the efficacy and burden of screening, as well as their perception of disease burden.

Women’s attitudes towards risk-reducing surgery for breast cancer were more variable than their attitudes towards risk-reducing surgery for ovarian cancer. For some women, the idea of waiting until a ‘lump’ was identified on a scan provided enough motivation to have an RRM.

*I was told I’d need to have scans and breast checks alternatively, six-monthly. Why would you do that for the rest of your life until they actually say, “Oh, yes, now we’ve found a lump”? I mean … that’s crazy to me, sitting on a time bomb going “Oh well, they’ll check me every six months and one day they might find a lump”. You’d say “why didn’t I just get that done at the start?” To me, the choice of having [breast] surgery was a no-brainer*. (Evie, 51 years, *BRCA2*)

Whereas most other women described breast cancer as more manageable due to being readily identifiable through screening or self-examination and *“unlikely to kill me”* (Elise, 55 years, *BRCA2*). This attitude is supported by the high proportion of women engaged in high-risk breast screening.

In contrast, nearly all women perceived their ovarian cancer risk as more threatening, or as Louise (56 years, *BRCA1*) described, “*a time bomb just waiting to go off*”. Similarly, Theresa described:

*[The doctor’s] summary was, “Well Theresa, you’d be pissed if you didn’t do anything and you got ovarian cancer, wouldn’t you?” And I thought, “That sums it up, doesn’t it?” It’s been brought to my attention and if removing the ovaries is a 100% certainty that you won’t get it… I’ve got an appointment in three weeks to see the gynaecological cancer specialist person about my ovaries*.(Theresa, 59 years, *BRCA2*)

Participants felt that undergoing RRM was “*pretty drastic” *(Elise, 55 years, *BRCA2*) compared to undergoing BSO, which was described as *“non-invasive”* (Liz, 60 years, *BRCA2*). Elise exemplified this logic when carefully weighing up her breast and ovarian cancer risk management options:

*I looked into [RRM] and thought it was pretty drastic and then the next option was the drug, the chemoprevention and so I looked into that as well and I weighed up the risks and decided not to do that. But I decided, yes, I would have my ovaries removed, because ovarian cancer is very hard to detect, whereas breast cancer is a slow growing and you can feel if there’s a lump, you can have a mammogram, and also, I was post-menopause. I didn’t really need my ovaries*.(Elise, 55 years, *BRCA2*)

With over half of the women with a *BRCA1/2* PV undergoing BSO, their pragmatism towards the BSO was evident in comments like: “*They [ovaries] did their job and yeah. They had to go*.” (Bridget, 54 years, *BRCA2*). Many were also menopausal or post-menopausal, which further justified their decision.

*Well, I was at that age [where] my ovaries were probably not working that well anyway because I was 55 … I’d started menopause so it was no big deal*.(Allison, 60 years, *BRCA2*)

### 3.4. Psychosocial Outcomes of Receiving Clinically Actionable Genetic Information

#### 3.4.1. Breast Cancer Risk Perception after Notification

Group 1 had significantly higher affective (*p* < 0.001) and cognitive (*p* < 0.001) breast cancer risk perception compared to group 2 (Table 3).

#### 3.4.2. Ovarian Cancer Risk Perception after Notification

Women who had received *BRCA1/2* genetic information from **life**pool had significantly higher affective ovarian cancer risk perception compared to group 2 (*p* = 0.0001), whereas there was no observable difference between groups for cognitive ovarian cancer risk perception (*p* = 0.1). However, when women who had a BSO were excluded, the remainder of group 1 had a significantly higher cognitive ovarian cancer risk perception (mean 35.5; SD 24.3) compared to group 2 (mean 22.7; SD 17.7; *p* = 0.02).

### 3.5. Cancer Worry after Notification

Despite recruitment being less than anticipated, a significant difference was identified in the CWS between the two groups, with group 1 having significantly higher cancer worry than group 2 (Cohen’s *d* = 0.64; *p* < 0.001) (Table 4). Furthermore, a greater proportion of group 1 (66.7%) met or exceeded the threshold for experiencing severe levels of worry about cancer occurrence or recurrence compared to group 2 (41.7%) (*p* = 0.01).

### 3.6. Women’s Reflections of Their Cancer Risk Perception and Cancer Worry

For many of the women from group 1, the link between their cancer risk perception and worry about developing cancer was evident. Many of these women could recall the numerical, absolute risk of breast or ovarian cancer associated with having a PV and ascribed different understandings of what it meant to them personally:

*The percentage of ovarian cancer [risk] is quite high if you are BRCA2 positive. It’s from 11% to 25%. So, it’s quite alarmingly high, one in four. “So, I don’t know whether I’m the one in four, you don’t know. It’s still something that worries you at the back of the mind”*.(Angela, 66 years, *BRCA2*)

After learning their genetic status, group 1 participants described becoming *“more attuned”* to their cancer risks as they had become *“a real thing”*, not just an abstract concept (Sharon, 75 years, *BRCA2*). The transition from knowing about cancer risks numerically to experiencing them subjectively was described by Sharon:

*“It just made me reassess … and even though I get it, even though you know what the [cancer] risk is in numbers, when you actually experience someone going through cancer, it does actually highlight the risk and make your perception of that risk a bit more attuned, I would say. So I think that that was like, “OK I don’t want to go through that.” This is a real thing not just an academic thing”*.(Sharon, 75 years, *BRCA2*)

Some group 1 participants shared how their heightened cancer risk perception manifested and exacerbated their cancer worry while waiting for risk-reducing surgery.

*“That plays on my mind a bit now, every twinge or any pain you get you’re like, “Oh, what if, what if, what if”, and I think I was like that pre getting my breasts removed. [It] was that every lump, everything would just play on my mind. I was always getting checked, and it was almost like a paranoia thing. But once they [my breasts] were gone, that was alleviated. I think once my ovaries are gone, a bit more of that anxiousness will be gone as well”*.(Lucy, 36 years, *BRCA2*)

### 3.7. Distress, Uncertainty, and Positive Experiences after Notification

Group 1 completed the MICRA to assess the specific impact of the disclosure of their genetic information conceptualised as distress, uncertainty, and positive experiences. Overall, these women had a total MICRA mean score of 27.4 (SD 11.8, range 9–56; possible range 0–95) (Table 5).

### 3.8. Adapting to Genetic Information Provided by **Life**pool

Group 1 completed the PAS to assess their adaptation to receiving the genetic information (Table 6). The mean PAS score was 2.9 (*SD* = 1.1), with 5 being the highest possible score.

### 3.9. Family Communication

All bar one participant in group 1 had informed their family members about the genetic information. Of the 41 group 1 respondents who were parents, 37 (90.2%) had told all their children the genetic information (another 2 had told some children; 4.9%). Of the 41 who had siblings, 35 (85.4%) had told all their brothers and sisters (and another 2 had told some siblings; 4.8%). The next most informed groups were nieces and nephews with 26 of 39 (66.7%) respondents informing all or some.

## 4. Discussion

The results from this study provide early evidence of the psychological and clinical outcomes for women who attended a population mammographic screening program, enrolled in research, and subsequently received unexpected genetic information about hereditary breast and ovarian cancer risk. A strength of this study was the recruitment of a matched comparator group, enabling the comparison of psychosocial outcomes for women who received this information and women who did not [18]. The women who received this information are among the first to receive cancer genetic information outside of the traditional models of care offered by CGS that includes pre-test genetic counselling [7]. Thus, the findings present the opportunity to examine whether the women who received information about a PV from a research cohort experienced outcomes that differ in comparison to women who receive this information through this traditional model of CGS.

Cancer worry and risk perception are frequently examined psychosocial outcomes after delivery of genetic information, interventions in CGS, or participation in cancer risk management programs [30,31]. These two outcomes are often reported as being correlated and are associated with subsequent decisions to engage in cancer risk management [32,33]. The case-control design of this study demonstrated that receiving clinically actionable genetic information increased women’s cancer worry and risk perception compared to women who did not receive this information [18]. Nevertheless, there is very little difference in cancer worry scores in the **life**pool population who received ‘genome-first’ care to populations who receive traditional pre-test genetic counselling and result disclosure in a CGS [30]. Eijzenga et al. (2015) tested a genetic counselling intervention and participants completed the CWS five months after receiving their genetic test results [30]. The mean CWS for the control group, who received standard genetic counselling and testing was 13.27 (95% CI 11.47–15.08), which is similar to group 1 in our study who received genetic information from **life**pool. In contrast, Chirico et al. (2021) studied women who had survived breast cancer, documenting their cancer worry 12 months after their primary treatment as substantially higher at 23.4 (SD 5.5) [34]. It is likely that the recency of these women’s breast cancer diagnoses is the primary reason for this much higher worry compared to the participants in our study. It is possible that the mode of notification employed by **life**pool incorporating telephone genetic counselling, followed by a referral to a CGS for genetic counselling and testing, works to mitigate any substantially detrimental experiences for participants [16]. Further, our findings illustrate how important timely and efficient access to CGSs are to confirm the research result, manage distress and promote adaptation by supporting participants’ engagement with cancer risk management strategies [10,35].

The previously unknown impact of receiving genetic information from a research study is gradually being elucidated from studies such as this and others that have returned clinically significant research results to participants. The participants in this study had a total MICRA score that was slightly higher (27.4, SD 11.8) compared to research participants in the high-risk melanoma group (24.2, SD 15.2) from Smit et al. (2018) [36]. The quantification of this impact can also be compared to patients who have received their genetic information through a clinical pathway. For example, Hamilton et al. (2021) examined the impact of receiving genetic results from a cancer gene panel for patients who had mainstream genetic testing [37]. While our participants reported slightly higher distress scores than identified by Hamilton at three weeks and three months after receipt of result, our participants had lower uncertainty compared to both time points and scored similarly for positive experiences.

Participants in our study reported that waiting to confirm if they were at high-risk of cancer was a source of distress, which prompted feelings of uncertainty. Health system responsiveness has previously been identified as a barrier to research participants taking up genomic information identified through research [38]. A model of service delivery that prioritises ‘genome-first’ care is required as research participants identified with a PV are highly likely to have this genetic information confirmed through clinical testing. Therefore, they can bypass many of the clinical processes in place for individuals with a personal/family history [7,10]. This may help mitigate distress and uncertainty upon receipt of this information.

How women adapt to this information is also under investigation. Adaptation, as defined by Biesecker et al. (2008), is “a dynamic and multi-dimensional process of coming to terms with the implications of a health threat and the outcomes of that process” [4,39]. Until the development and validation of the Psychological Adaptation Scale, no metric existed that comprehensively measured adaptation to genetic information, and therefore no cross-study comparisons were available [4]. Shapira et al. (2018) used the PAS to measure the adaptation of women with a *BRCA1/**2* PV finding a mean of 3.04 (SD 1.11) [40] compared to women in our study with a mean PAS of 2.9 (SD 1.1). Shapira et al. also noted that women in their study who had a risk-reducing mastectomy had higher adaptation scores compared to women who had not [40].

Overall, results from our study provide preliminary evidence that receiving unexpected genetic information through research participation can be distressing although not obviously different from receiving similar information from CGS.

### Limitations

A limitation of this study is the women who attended **life**pool and subsequently participated in this research could be viewed as proactive with regards to health information as they attended a population-based breast screening program, consented to research, and opted in to receive unexpected genomic information. We did not have access to women who do not attend a population-based screening program or those who are less inclined to participate in research. These individuals may have different outcomes. The views of men were also not captured. 

## 5. Conclusions

Our findings demonstrate that the clinical and psychosocial outcomes of individuals who receive cancer risk information through ‘genome-first’ care are not dissimilar to those reported for patients who attend CGS. This alternative avenue of identifying individuals and their families in the population who have a significantly increased risk of cancer does not pose unmanageable risks due to the mechanism of notification involving telephone genetic counselling and follow up genetic counselling through a clinical service. A model of genetic counselling for ‘genome-first’ care is needed that incorporates expedited access to CGS to mitigate feelings of anxiety and uncertainty by confirming participants’ high-risk of cancer and enabling engagement with cancer risk management strategies.

## Figures and Tables

**Table 1 jpm-12-01112-t001:** Demographic characteristics comparing groups 1 and 2.

Demographic	Total	Group 1 PV	Group 2 No PV	*p* Value
Age (years)	*n* = 138	*n* = 45	*n* = 93	
Mean (SD)	64.4 (6.7)	65.2 (8.2)	64.1 (5.8)	0.4
Range	37–79	37–79	48–77	
Education	*n* = 140	*n* = 44	*n* = 96	
Non-tertiary	86 (61.4)	26 (59.1)	60 (62.5)	0.7
Tertiary	54 (38.6)	18 (40.9)	36 (37.5)	
Income (per annum)	*n* = 140	*n* = 44	*n* = 96	
<$90,000AUD	83 (59.3)	23 (52.3)	60 (62.5)	0.3
>$90,000AUD	57 (40.7)	21 (47.7)	36 (37.5)	
Relationship	*n* = 141	*n* = 45	*n* = 96	
Partnered	118 (83.7)	39 (86.7)	79 (82.3)	0.5
Not partnered	23 (16.3)	6 (13.3)	17 (17.7)	
Children	*n* = 141	*n* = 45	*n* = 96	
Yes	118 (83.7)	42 (93.3)	76 (79.2)	0.03 *
No	23 (16.3)	3 (6.7)	20 (20.8)	
Pathogenic variant	*n* = 141	*n* = 45	*n* = 96	
Not detected	96 (68.1)	0 (0)	96 (100)	
*ATM*	1 (0.7)	1 (2.2)	0 (0)	
*BRCA1*	2 (1.4)	2 (4.4)	0 (0)	
*BRCA2*	24 (17.0)	24 (53.3)	0 (0)	
*PALB2*	15 (10.6)	15 (33.3)	0 (0)	
HBOC predisposition/Cannot recall	3 (2.1) ^a^	3 (6.6) ^a^	0 (0)	
Mean time since notification of genetic information (years)	*n* = 141	*n* = 45	*n* = 96	
Mean (SD)	n/a	1.7 (1.5)	n/a	
Range	n/a	0.2–5.1	n/a	
Cancer history	*n* = 141	*n* = 45	*n* = 96	
No cancer	126 (89.4)	30 (66.7)	96 (100)	
Breast cancer	9 (6.4)	9 (20.0)	0 (0)	
Ovarian cancer	1 (0.7)	1 (2.2)	0 (0)	
Other cancer	5 3.6)	5 (11.1)	0 (0)	

^a^ Two had not had confirmatory clinical genetic testing at the time of the study and one could not specifically recall her result; * Denotes statistically significant result (*p* < 0.05).

**Table 2 jpm-12-01112-t002:** Clinical outcomes after notification of genetic information.

Clinical Outcomes	*n* (%)
Confirmatory genetic testing	*n* = 45
Complete	43 (95.6)
Waiting for FCC appointment	1 (2.2)
Undecided	1 (2.2)
Uptake of breast cancer risk management strategies	*n* = 45
High-risk breast screening	27 (60.0)
Chemoprevention ^a^	10 (22.2)
Risk-reducing mastectomy	6 (13.3)
Uptake of ovarian cancer risk management strategies	*n* = 26 ^b^
Bilateral salpingo-oophorectomy	14 (53.9)

^a^ 9/10 participants taking tamoxifen reported they were also having breast screening; ^b^ Participants with a *BRCA1/2* PV only.

**Table 3 jpm-12-01112-t003:** Comparison of breast and ovarian cancer risk perception between groups 1 and 2.

Breast Cancer Risk Perception				
	Total	Group 1 PV	Group 2 No PV	*p* Value
Affective (0–3)	*n* = 141	*n* = 45	*n* = 96	
Range	0–3	0–3	0–3	Cohen’s *d* = 1.13 t = 6.2 (df 139) *p* < 0.001 *
Mean (SD)	0.9 (0.8)	1.4 (0.8)	0.6 (0.6)	
95% CI	0.7–1.0	1.1–1.6	0.5–0.7	
Cognitive (0–100)	*n* = 141	*n* = 45	*n* = 96	
Range	0–80	0–80	0–80	Cohen’s *d* = 0.85 t = 5.0 (df 139) *p* < 0.001 *
Mean (SD)	31.6 (21.6)	43.8 (23.5)	25.9 (18.1)	
95% CI	28.0–35.2	36.7–50.9	22.2–29.5	
**Ovarian Cancer Risk Perception**				
	**Total**	**Group 1** **PV**	**Group 2** **No PV**	***p* Value**
Affective (0–3)	*n* = 125	*n* = 29	*n* = 96	
Range	0–3	0–3	0–2	Cohen’s *d* = 0.68 t = 4.1 (df 123) *p* < 0.001 *
Mean (SD)	0.6 (0.8)	1.1 (1.1)	0.5 (0.6)	
95% CI	0.5–0.8	0.7–1.6	0.4–0.6	
Cognitive (0–100)	*n* = 125	*n* = 29	*n* = 96	
Range	0–100	0–100	0–70	Cohen’s *d* = 0.28 t = 1.6 (df 123) *p* = 0.1
Mean (SD)	24.4 (22.7)	30.3 (34.3)	22.7 (17.7)	
95% CI	20.4–28.5	17.2–43.3	19.1–26.3	

* Denotes statistically significant result (*p* < 0.05).

**Table 4 jpm-12-01112-t004:** Comparison of cancer worry between groups 1 and 2.

	Total	Group 1 PV	Group 2 No PV	*p* Value
	*n* = 141	*n* = 45	*n* = 96	
Range	8–25	10–25	8–22	<0.001 *
Mean (SD)	13.6 (3.0)	14.9 (3.1)	13.1 (2.7)	
95% CI	13.2–14.1	13.9–15.8	12.5–13.6	
High cancer worry	70 (49.7)	30 (66.7)	40 (41.7)	0.01 *
Low cancer worry	71 (50.4)	15 (33.3)	56 (58.3)	

* Denotes statistically significant result (*p* < 0.05).

**Table 5 jpm-12-01112-t005:** Women’s distress, uncertainty, and positive experiences of receiving clinically actionable genetic information (MICRA).

MICRA Sub-Scale	Mean (SD)	Reported Range
Distress (0–30)	7.2 (6.9)	0–26
Uncertainty (0–45)	9.8 (6.0)	0–24
Positive experiences (0–20)	7.3 (4.9)	0–20
TOTAL (0–95)	27.4 (11.8)	9–56

**Table 6 jpm-12-01112-t006:** Women’s adaptation to receiving clinically actionable genetic information from **life**pool.

PAS Sub-Scale	Mean (SD)	Reported Range
Coping efficacy (4–20)	12.9 (4.4)	4–20
Self-esteem (5–25)	13.7 (6.0)	5–25
Social integration (3–15)	8.2 (4.0)	3–15
Spiritual wellbeing (3–15)	8.8 (4.2)	3–15
Total PAS (15–75)	43.6 (16.5)	15–75
Adaptation score	2.9 (1.1)	1–5

## Data Availability

The data presented in this study are not publicly available due to restrictions on maintaining participants’ privacy.

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
