# Peer review of "The Clinical and Psychosocial Outcomes for Women Who Received Unexpected Clinically Actionable Germline Information Identified through Research: An Exploratory Sequential Mixed-Methods Comparative Study"

_jpm, 2022, doi:10.3390/jpm12071112_

Round 1

Reviewer 1 Report

This manuscript addresses a very interesting topic and sometimes a little forgotten by health personnel who provide genetic counseling to high-risk patients, which is the evaluation of their phycosocial health. Therefore I highly recommend it for publication with the following improvements

1.- All gene acronyms must be described somewhere in the text and in italics (this includes tables).

2.-In the present study the authors only included patients with a pathogenic variant (PV), they did not have patients with variants of uncertain significance (VUS)?, and if they did, why were they not included? In our experience we have observed that patients who are given this result also generate high levels of stress and anxiety.

3.-I consider that all statistically significant P values ​​in the tables should have an asterisk 

6.-Some patient testimonials are in italics and others are not, please homogenize so that everything has the same format. 

7.- For ethical reasons, I consider that it is better that the name of the patient does not appear in her testimony. Better place as Patient 1, 2, 3, etc.

8.-In paragraph three of the discussion there is a statement with another type of letter, please homogenize

9.-I believe that within the discussion they should include perspectives of the study.

10.-I am left wondering if the patients who came out with a high score for the MICRA test had psychological support from the center

Author Response

  1. All gene acronyms must be described somewhere in the text and in italics (this includes tables).

Genes have been described in text and in tables using their symbol as prescribed by the HUGO Gene Nomenclature Committee (https://www.genenames.org/about/guidelines/) and are now consistently italicised throughout.

  1. In the present study the authors only included patients with a pathogenic variant (PV), they did not have patients with variants of uncertain significance (VUS)?, and if they did, why were they not included? In our experience we have observed that patients who are given this result also generate high levels of stress and anxiety.

VUS identified by lifepool were not returned to participants because they were not considered to be clinically actionable. Therefore, this population was not available to be included in our study.

  1. I consider that all statistically significant P values ​​in the tables should have an asterisk

We have marked all p values <0.05 in the tables with an asterisk.

  1. Some patient testimonials are in italics and others are not, please homogenize so that everything has the same format.

We have ensured all participant quotes are in italics.

  1. For ethical reasons, I consider that it is better that the name of the patient does not appear in her testimony. Better place as Patient 1, 2, 3, etc.

As described on page 6 under the subheader ‘Interviews’: “Transcripts were quality controlled, de-identified, and pseudonyms assigned before analysis.”

For clarity, we have added further information to pg 7 under the subheader ‘Qualitative’: “Participant quotes are used to illustrate the findings with pseudonym, age at interview in years, and the gene within which the germline PV was identified in parentheses.”

  1. In paragraph three of the discussion there is a statement with another type of letter, please homogenize

Paragraph 3 in the discussion has been substantially re-written and no longer mentions a letter.

  1. I believe that within the discussion they should include perspectives of the study.

We have re-written some of the discussion and hope that it now better addresses the perspective of this study.

  1. I am left wondering if the patients who came out with a high score for the MICRA test had psychological support from the center

None of the participants met the criteria stipulated in our protocol to manage distress. As there is no validated clinical cut off for the MICRA, we did not use it to screen participants for distress.

Reviewer 2 Report

This is an interesting paper assessing the effects of ‘genome first’ cancer genetic testing results. This is an important area to explore but the presentation is somewhat muddied, as noted below.

Major points

·         The introduction of this paper (and abstract) focuses on comparing genome first results to the CGS model, but the results section only compares women who received results to those who did not, and only the discussion compares to the CGS model. I suggest either presenting the comparison to CGS in the results or re-framing the focus of the paper/introduction.

Minor points

·         Inconsistent use of bolding in lifepool- sometime life is bolded, other times it is not

·         Pg 6: final quote from Evie not in italics like the others

·         Pg 9: quote from Angela and Lucy not in italics, name split between rows

·         Why was cognitive ovarian cancer risk perception similar between the groups? Did this change if cases that underwent BSO cases were excluded?

·         Discussion feels like a disjointed set of comparisons- as noted above, probably fits better into results.

Author Response

This is an interesting paper assessing the effects of ‘genome first’ cancer genetic testing results. This is an important area to explore but the presentation is somewhat muddied, as noted below.

Major points

The introduction of this paper (and abstract) focuses on comparing genome first results to the CGS model, but the results section only compares women who received results to those who did not, and only the discussion compares to the CGS model. I suggest either presenting the comparison to CGS in the results or re-framing the focus of the paper/introduction.

We have re-written parts of the introduction to clarify the focus of the paper.

Minor points

Inconsistent use of bolding in lifepool- sometime life is bolded, other times it is not

We have made sure all mentions of lifepool are consistently formatted.

Pg 6: final quote from Evie not in italics like the others

We have corrected this.

Pg 9: quote from Angela and Lucy not in italics, name split between rows

We have corrected this.

Why was cognitive ovarian cancer risk perception similar between the groups? Did this change if cases that underwent BSO cases were excluded?

We suspect that the inclusion of the 14 women who had a BSO in this comparison of risk perception was the reason that cognitive ovarian cancer risk perception was similar between groups. We examined this outcome again excluding women who had a BSO and found a difference in risk perception. This finding has been added to the results on pg 11 under the subheader ‘Ovarian cancer risk perception’.

Discussion feels like a disjointed set of comparisons- as noted above, probably fits better into results.

We have re-written some of the discussion and hope that it now better fits with the findings of this study.